# Evaluating the Health of Urban Human Settlements

Chunmei Zhang [1] and Lingen Wang [2],*

1    School of Government, Liaoning Normal University, Dalian 116029, China
2    Institute of Geographic Sciences and Natural Resources Research, Chinese Academy of Sciences, Beijing 100101, China
*    Correspondence: wangle@igsnrr.ac.cn

**Abstract:** The design and dynamics of a human settlement affect the health of its residents; for example, high-quality human settlements can improve the health of their residents. Accordingly, it is important to evaluate and improve the quality of human settlements, especially from a health perspective. Taking on this task, this study applied an entropy method and spatial autocorrelation analysis to evaluate the human settlement quality of 14 prefecture-level cities in Liaoning Province. The results provided the following three main insights. (1) The settlements were of ordinary quality and effective measures should be taken to improve their quality. (2) Regarding spatial characteristics, from 2009 to 2019, these settlements showed clear regional differentiation, with a low spatial distribution in the east, high spatial distribution in the west, high spatial distribution in the middle, and slightly lower spatial distribution at both ends. These characteristics reflect imbalances in the quality of the regional human settlements. (3) Regarding subsystem evolution characteristics, the quality of the settlements showed clear systematic differentiation during the evaluation period. In response to these findings, this paper proposes effective measures to improve the quality of urban human settlements and provides theoretical support for the healthy development of such settlements, including the revitalization and development of old industrial bases.

**Keywords:** human habitation; degree of health; Healthy China; Liaoning Province; healthy city

## 1. Introduction

Urban human settlements are closely related to human health and happiness. The rapid expansion of cities and populations as a result of economic development has brought considerable challenges for human settlements, especially regarding public health [1,2]. For example, mega-scale urbanization and industrialization have triggered a series of health crises while changing the spatial forms of cities [3–6], evident in the trends for taller buildings, greater building density, and increasingly smaller living spaces, which all pose a threat to human health. According to reports, the urban areas of cities have been most affected by the health challenges posed by the COVID-19 pandemic [7]. The 17 sustainable development goals of the United Nations include good health and well-being as well as sustainable cities and communities. A healthy living environment contributes to the realization of human health and well-being, and thus the realization of human sustainable development. China's 19th Party Congress elevated the construction of a healthy China to a national priority development strategy, emphasizing the relationship between health and the environment [8,9]. Specifically, the Healthy China 2030 initiative calls for "integrating health into the whole process of urban and rural planning, construction, and governance, and promoting the coordinated development of cities and people's health" [10]. In light of this initiative, it is of great importance to evaluate the quality of urban human settlements in China and determine effective measures for improving their health quality.

Scholars have long been concerned with healthy living conditions. Notably, Howard's Garden City theory reflects the desire for a healthy living environment [11]. Meanwhile, in 1843, Edwin Chadwick noted that the poor living conditions of the working class were a

source of disease outbreaks in his "Report on the Environmental Health of the Working Class in England" [12]; in the 19th century, urban planning sought to curb the outbreak of infectious diseases by improving housing conditions [13]. In the 1980s, the World Health Organization (WHO) proposed the concept of a "healthy city"; this stimulated widespread interest in this issue, prompting people to reflect on how urban human settlements impact public health [14]. At present, the health of urban human settlements has become a focus of research. Studies have analyzed the relationship between urban human settlements and health from the perspectives of planning, architecture, health care, and geography, among others [15–21], and have sought to derive planning and construction strategies for healthy human settlements [22–24]. Based on the existing research results, the need to construct healthy human settlements has been recognized by the academic community. However, the construction of healthy human settlements requires an effective system to evaluate, guide, and correct the construction process. Some work has already been done in this regard; specifically, Luo et al. [25] developed a spatial evaluation index of the health of small and medium-sized cities, Barton et al. [26] used ecological methods to evaluate the factors of human settlements that affect human health, and Yang et al. [27] used spatial indicators to evaluate the health of urban environments in Shenzhen, China.

While existing studies serve as important references for the current study, the indexes constructed in previous research primarily focus on how living environments harm resident health (e.g., energy consumption per 10,000 yuan of gross domestic product [GDP] and per capita industrial sulfur dioxide emissions) and health-friendly safeguards for living environments (e.g., the number of hospital beds per 10,000 people and environmental protection investment as a proportion of GDP). In contrast, scholars have rarely considered how living environments can encourage healthy behavior. Notably, the WHO believes that a healthy city can help people live healthier and longer lives with less disease [28]. An important recognition of healthy cities is that traditional public health approaches focused on preventing or treating disease cannot adequately address new health risks and eradicate the underlying causes of these health risks [29]. The Healthy China strategy emphasizes that health is rooted in the improvement of health ability and thus that the construction of healthy urban human settlements should be based on optimizing space—for example, settlements should include public spaces conducive to healthy lifestyles—and minimizing the health risks of rapid urbanization [23,30].

This study conducted a case study of 14 prefecture-level cities in Liaoning Province, China with an entropy method using geographic information technology. We sought to identify effective measures to improve the quality of urban human settlements and to provide theoretical support for best practices for the healthy development of such settlements, including the revitalization and development of old industrial bases. This study notably contributes to research on the topic of urban human settlements by constructing an index to evaluate the health quality of urban human settlements and offering insights into the current situation in Liaoning Province and the best practices for optimizing urban human settlements there.

## 2. Materials and Methods

### 2.1. Study Area

The study area included 14 prefecture-level cities in Liaoning Province. The geographical location of the study area is shown in Figure 1. An old Chinese industrial base, Liaoning Province is highly urbanized and industrialized. Indeed, its urban landscape is a microcosm of China. Therefore, research on the levels of health in urban human settlements in Liaoning Province has practical significance for improving the health quality of human settlements in the context of rapid urbanization.

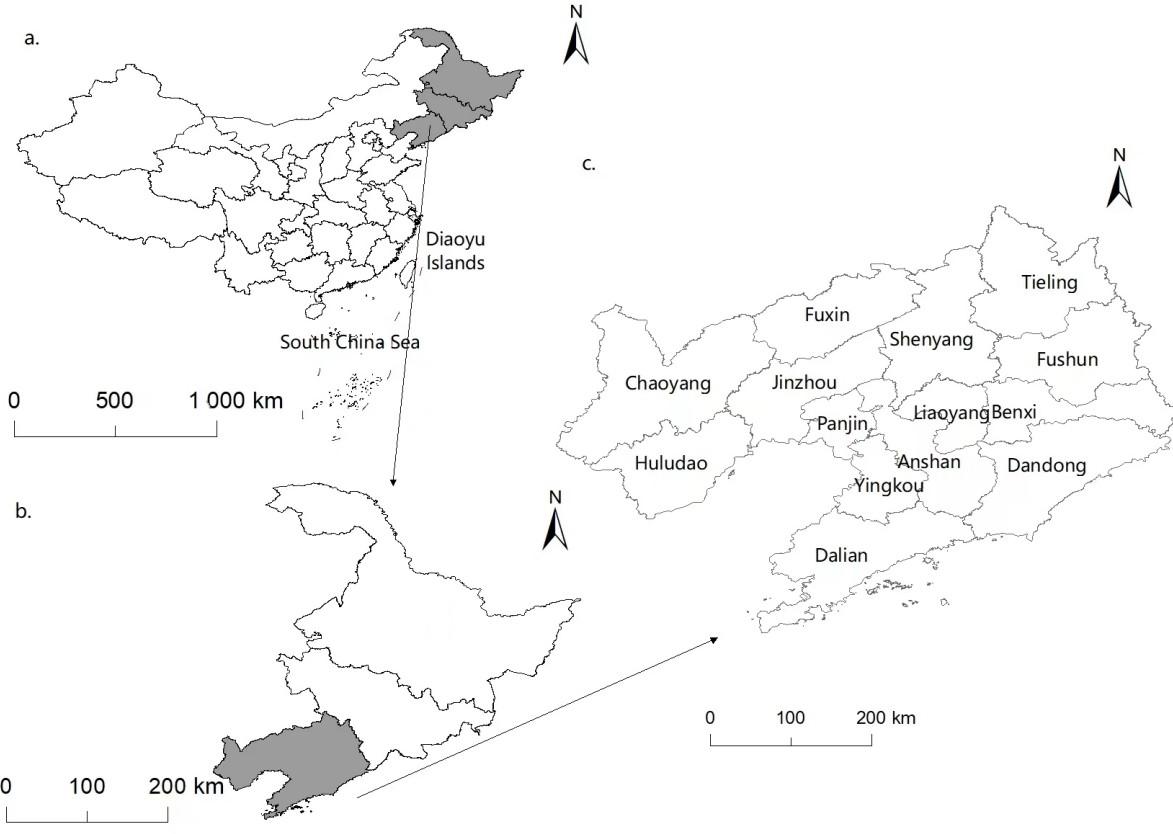

**Figure 1.** The geographical location of the study area: (**a**) China, (**b**) Liaoning, (**c**) The 14 cities in Liaoning in this case study.

## 2.2. Evaluation System and Data Sources

The WHO believes that a healthy city should be an organic whole comprising healthy people, a healthy environment, and a healthy society [31]; a healthy human settlement environment should also include these elements. Based on the WHO's concept, we constructed first-level indicators for five dimensions: population, housing, social, environmental, and facility systems. Drawing on the concept of a "healthy China," we investigated the quality of each of the five constituent systems of the human settlement environment in terms of health risk, health protection, and health promotion [23,30]. "Health risk" refers to factors in human settlements that are detrimental or that pose dangers to the health of both the system and the residents, such as air pollution [32–34]. Further, a health risk can also comprise the accumulation or imbalance of certain factors that can put pressure on urban human settlements, such as the aging rate (e.g., an aging society can inhibit sustainable population development). In contrast, "health protection" refers to factors in human settlements that can protect the system and the health of residents, such as the number of hospital beds per 10,000 people and the safety of the waste treatment system [35,36]. Last, "health promotion" refers to factors in human settlements that can encourage healthy lifestyles [37], such as the settlement's active intervention in and guidance to support resident health (e.g., by optimizing per capita public green space area, the number of public libraries per 100 people, and the ratio of education expenditures to GDP). Finally, 39 indicators were selected based on the spirit of the Healthy China initiative and the availability of data to build an urban human settlement health evaluation index system. The comprehensiveness and operability of the index system were considered based on previous research [38–40]. The index system is shown in Table 1.

**Table 1.** Evaluation index system for the health of human settlements.

| Standard Layer | Evaluation Dimension | Index Layer |
| --- | --- | --- |
| Population system | Health risk | Mortality rate (%)<br>Proportion of urban population with the lowest social security (%)<br>Aging rate (%) |
| | Health protection | Proportion of working-age population (%) |
| | Health promotion | Natural population growth rate (%)<br>Proportion of teenagers in the population (%)<br>Birth rate (%)<br>Number of college students per 10,000 people (persons) |
| Residential system | Health risk | Population density (persons)<br>Housing price index (%) |
| | Health protection | Per capita living space ($m^2$)<br>Housing security investment (yuan)<br>Comprehensive service facilities in urban communities |
| | Health promotion | Proportion of housing investment in GDP (%) |
| Social system | Health risk | Number of criminal cases<br>Unemployment rate (%) |
| | Health protection | Per capita disposable income of urban residents (yuan)<br>Participation of urban and rural residents in basic old-age insurance (persons)<br>Medical insurance participation (persons) |
| | Health promotion | Proportion of expenditure on education in GDP (%)<br>Proportion of scientific and technological expenditure in GDP (%) |
| Environmental system | Health risk | Industrial wastewater emissions (10kt)<br>Industrial sulfur dioxide emissions (t)<br>Industrial dust emissions (t) |
| | Health protection | Centralized treatment rate of urban domestic sewage (%)<br>Air quality excellent rate (%)<br>Harmless treatment rate of garbage (%) |
| | Health promotion | Per capita green space ($m^2$) ratio in constructed areas with public green space |
| Facility system | Health risk | Hazardous waste storage (t)<br>Per capita domestic water consumption of residents (t) |
| | Health protection | Number of public transportations per 10,000 people<br>Number of hospital beds per 10,000 people<br>Per capita urban road area ($m^2$)<br>Number of health institutions |
| | Health promotion | Number of centers for disease control and prevention<br>Public library collection per 100 people (number)<br>Number of stadiums and gymnasiums<br>Internet penetration rate (%) |

The index data were derived from the statistical yearbooks for Liaoning Province from 2010 to 2021. Due to differences in the quality of the statistics, some data were revised and adjusted using statistical methods, referring to the Statistical Bulletin of the National Economic and Social Development of each city. Specifically, missing data were filled with the average growth rates since 2005. The data acquisition in this study followed the principles of authority and reliability, which can truly reflect the quality of urban human settlements in Liaoning Province.

*2.3. Methods*

This study applied an entropy method, spatial autocorrelation analysis, and multiple linear regression analysis to evaluate the human settlement quality of 14 prefecture-level cities in Liaoning Province. These methods are detailed below.

Entropy method: To overcome the interference of subjective factors in determining the weight of indexes, the entropy method was used to weigh each index [41]. According to the definition of "information entropy," the entropy value can be used to judge the dispersion degree of an indicator. The smaller the information entropy value, the greater the dispersion degree of the indicator and the greater the impact of the indicator on the comprehensive evaluation (i.e., the weight). Thus, the weighted summation method was used to measure and evaluate the degree of health in urban human settlements in Liaoning Province, and is represented as follows:

(1)    Standardize the data to eliminate the differences caused by different dimensions.
(2)    Calculate the proportion of *j* indicators in the *i* region; *m* is the number of regions:

$$P_{ij} = \frac{x_{ij}}{\sum_{i=1}^{m} x_{ij}} \tag{1}$$

(3)    Calculate the entropy value of index *j*; *m* is the number of regions:

$$e_j = -k \sum_{i=1}^{m} (p_{ij} \, ln \, p_{ij}), k = 1/ln \, m, e_j \in [0,1] \tag{2}$$

(4)    Calculate the weight of index *j*:

$$w_j = \frac{1 - e_j}{\sum_{j=1}^{m} (1 - e_j)} \tag{3}$$

(5)    Calculate the final score of human settlements; *n* is the number of indicators:

$$V = \sum_{j=1}^{n} w_j \times x_{ij} \tag{4}$$

Spatial autocorrelation analysis: Spatial autocorrelation methods can be roughly divided into two categories according to their functions: global autocorrelation and local autocorrelation [42–44]. Global spatial autocorrelation describes the spatial characteristics of a phenomenon in the whole region and measures the degree of agglomeration in areas with similar human settlements. Moran's I coefficient is most commonly used to measure this autocorrelation [44] and is represented as follows:

$$I = \frac{n}{s} \frac{\sum_{i=1}^{n} \sum_{j=1}^{n} W_{ij}(y_i - \overline{y})(y_j - \overline{y})}{\sum_{i=1}^{n}(y_i - \overline{y})^2} \tag{5}$$

$W_{ij}$ is the spatial weight, which represents the location relationship between region *i* and region *j*; *n* Is the number of observations; $y_i$ and $y_j$ represent the observed values of sample *i* and *j*; $\overline{y}$ is the average value of the sample points.

Multiple linear regression: In a real society, the emergence of a phenomenon is often not the result of a single factor, but rather a result of the joint action of multiple factors. A human settlement is a giant complex system, and its influencing factors are also characterized by pluralism and complexity [45–47]. As such, changes in dependent variables must be reflected through multiple independent variables. Therefore, this study used multiple linear regression to study the factors affecting differences in the environmental quality of the human settlements across Liaoning Province.

## 3. Results

### *3.1. Evolution Characteristics of Human Settlement Quality*

### 3.1.1. Temporal Evolution Characteristics

The entropy method (Formulas (1)–(3)) is used to calculate the weight of the evaluation index of the human settlements, and the evaluation of the overall health quality of residential areas in Liaoning Province is obtained through Formula (4). This study's comparative analysis of the health quality of urban human settlements in Liaoning Province between 2009 and 2020 revealed the following findings. (1) The settlements had an ordinary overall level of health quality (average value of 0.4809). Nine cities, accounting for 64.29% of the total number of cities in the province, had below-average values. Meanwhile, Dalian and Shenyang had relatively high values at 0.6312 and 0.6237, respectively. These higher values were found early in the evaluation period. (2) As Figure 2 shows, based on the average value of each time period, the urban human settlement health index in Liaoning Province fluctuated, showing an "M"-shaped evolution trend, accompanied by a downward trend in a rising process of fluctuation. While the index declined in some years between 2009 and 2017, it generally demonstrated an upward trend. From 2017 to 2020 the index dropped significantly from 0.493 to 0.461, a decrease of 6.5%. These findings show that urban human settlement quality in Liaoning Province did not demonstrate a positive trend during this period of social and economic development.

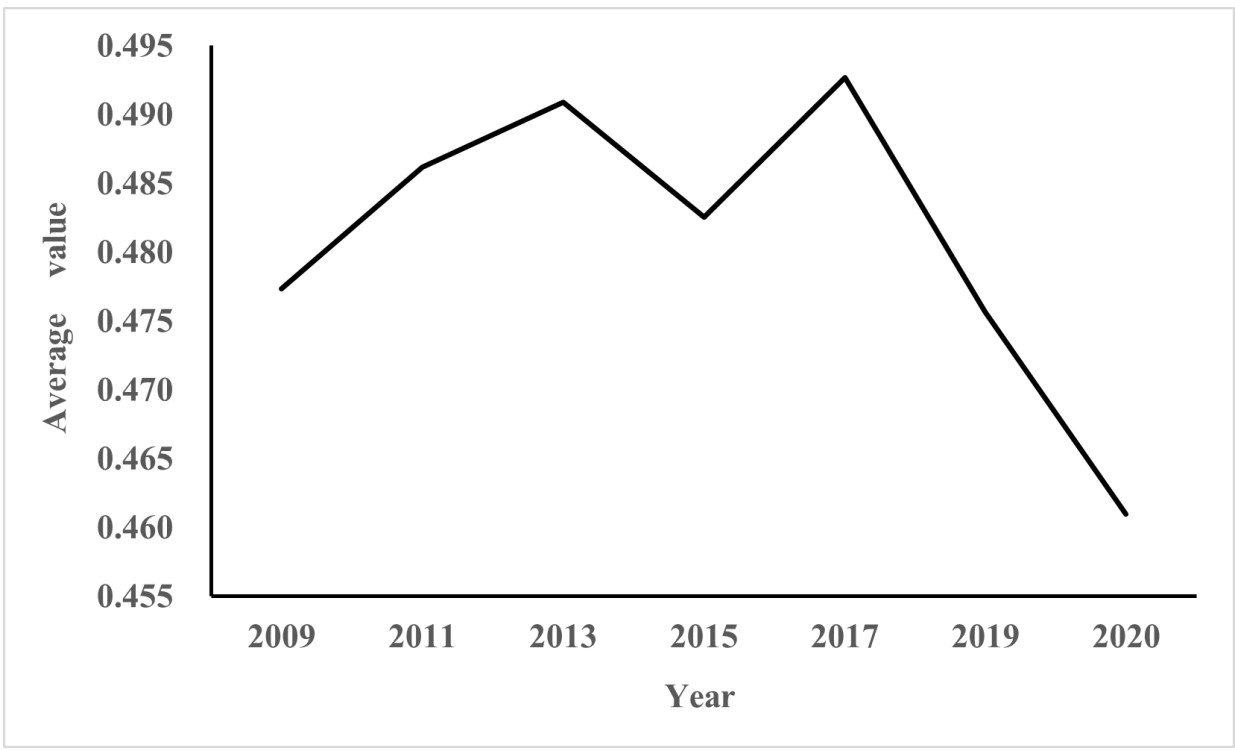

**Figure 2.** Average value of the degree of health of urban human settlements in Liaoning Province from 2009 to 2020.

### 3.1.2. Analysis of Spatial Evolution Characteristics

The natural breakpoint method was employed to grade the urban human settlements' qualities and subsystems in the study area during the evaluation period using ArcGIS10.2. Quality grades were divided into five levels: high, relatively high, medium, relatively low, and low. The spatial distribution pattern of the quality of urban human settlements in Liaoning Province from 2009 to 2020 is shown in Figure 3.

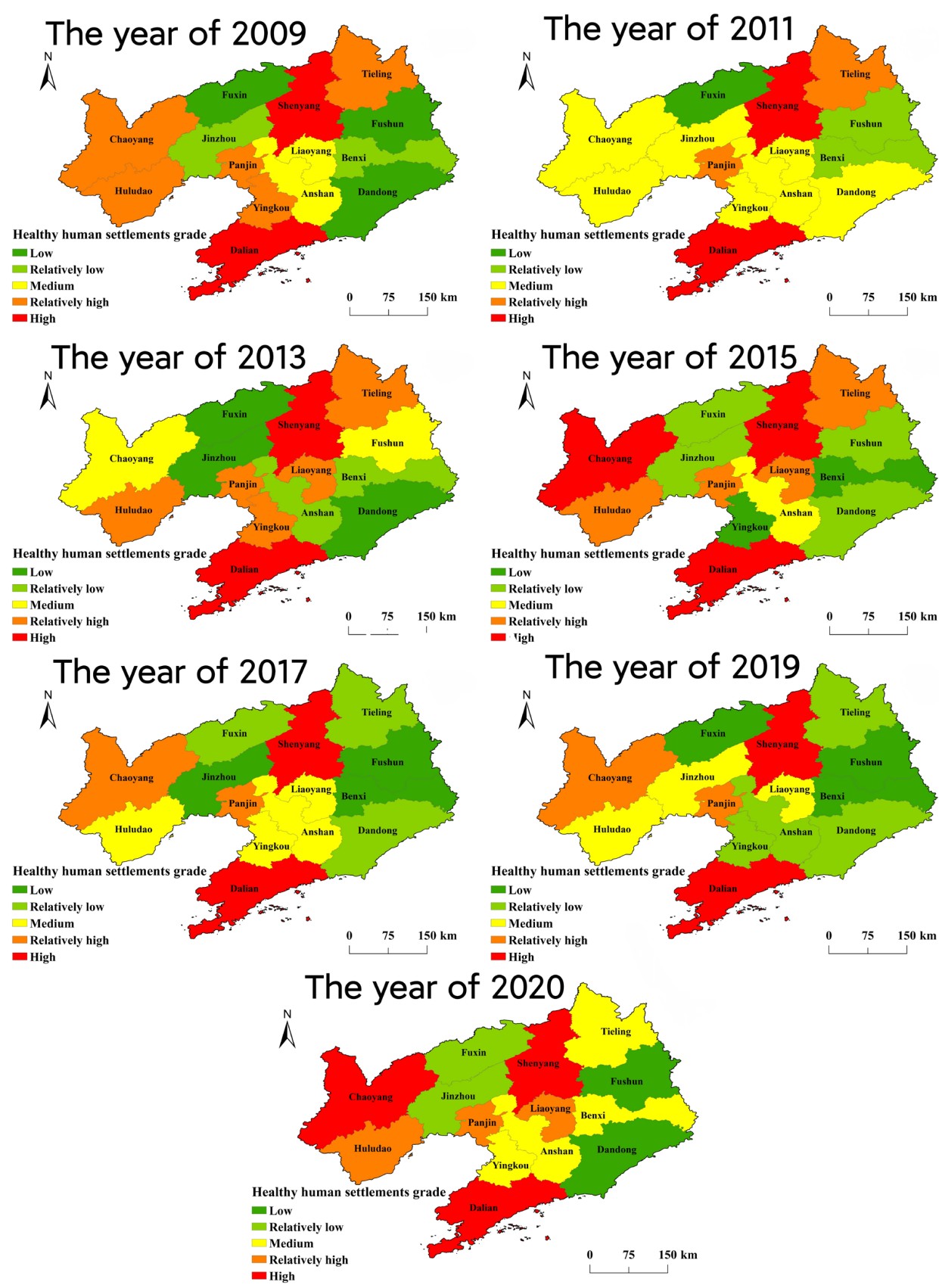

**Figure 3.** Spatial distribution pattern of the level of health in the urban human settlements in Liaoning Province from 2009 to 2020.

During these 11 years, the quality of these urban human settlements exhibited distinctive regional differentiation, forming a "core-edge" pattern and showing a spatial distribution pattern of attenuation from the core area to the edge area as a whole. Most of the high-value areas were concentrated in Shenyang, Dalian, and surrounding cities. The cities in the low-value areas were relatively solidified, showing a spatial distribution trend of lower values in the east, higher values in the west, a raised spatial distribution in the middle, and slightly lower values at both ends. The average value in southern Liaoning (Dalian and Yingkou) was the highest, followed by those in northern Liaoning (Shenyang, Tieling, and Fushun), western Liaoning (Jinzhou, Huludao, Panjin, Chaoyang, and Fuxin), central Liaoning (Anshan and Liaoyang), and eastern Liaoning (Dandong and Benxi) The spatial correlation of the quality of urban human settlements was weak. Equation (5) was used to calculate the global Moran's I estimate of the distribution of urban human settlement health in 2009, 2011, 2013, 2015, 2017, 2019 and 2020. As shown in Table 2, during the evaluation period, the p-values were greater than 0.05 and the Z values of the normal statistics were less than 1.96; they thus failed to pass a significance test, indicating that the spatial correlation of the health of the urban human settlements was weak. Therefore, in the context of the Healthy China initiative and new urbanization, cities and regions should connect and cooperate closely with each other to optimize health. It is also necessary to improve the quality of human settlements in marginal cities using the driving role of core cities to build a healthy Liaoning.

**Table 2.** Global Moran's I estimate of the health of human settlements.

| Year | Moran's I | Z (I) | *p* |
| --- | --- | --- | --- |
| 2009 | 0.105 | 0.664 | 0.507 |
| 2011 | −0.027 | 0.196 | 0.843 |
| 2013 | 0.168 | 0.918 | 0.359 |
| 2015 | 0.049 | 0.448 | 0.654 |
| 2017 | 0.374 | 1.627 | 0.104 |
| 2019 | 0.162 | 0.860 | 0.390 |
| 2020 | 0.028 | 0.397 | 0.691 |

### 3.1.3. Analysis of Subsystem Evolution Characteristics

The comparison of the evaluation results of the five systems revealed distinctive temporal and spatial differences in the index patterns of the social, human, facilities, environmental, and residential systems.

Systematic temporal differentiation: (1) System time differentiation. As shown in Figure 4. During the study period, there were certain differences in the change trends in the five major systems of urban human settlements in Liaoning Province. (2) The overall environmental system and facility system showed a trend of fluctuation and increase, indicating that the urban environment and infrastructure construction in Liaoning Province had improved overall and improved to a certain extent from 2009 to 2020. (3) The human system, social system and residential systems have declined to a certain extent. The decline of the residential system was small. The downward trend of the human system was obvious, from 0.523 in 2009 to 0.459 in 2019. The main reason is that the natural population growth rate and birth rate have declined significantly, and the aging rate has continued to rise. During the evaluation period, the degree of aging in 14 prefecture level cities has continued to increase, and the trend in population structure change deserves attention. The social system declined from 0.420 in 2009 to 0.389 in 2020, mainly due to the impact of downward pressure on the economy, and some cities' scores on indicators such as the unemployment rate and the proportion of education expenditure decreased to a certain extent.

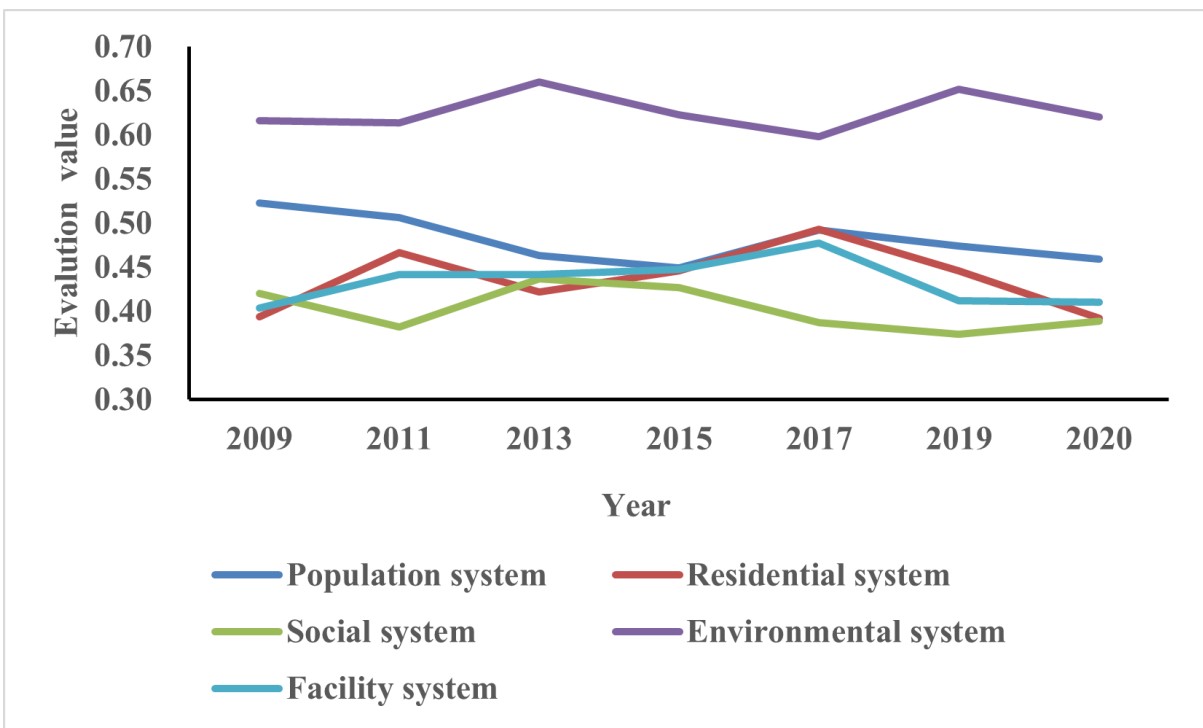

**Figure 4.** Temporal differentiation of the five systems.

The human settlement environment is composed of five systems. Only the coordinated development of the five systems can realize the continuous improvement of the quality of human settlement environment. The evaluation found that the quality of the human system and social system has declined significantly. It is necessary to optimize the fertility policy and improve the talent policy to cope with the declining proportion of the workforce. It is additionally necessary to take effective measures, such as reforming state-owned enterprises and promoting urban economic development.

Spatial differentiation of the system: As shown in Figure 5. The difference between the population system's maximum and minimum scores was 0.364, with a standard deviation of 0.123. It is worth mentioning that there was a significant difference in the population system among cities, and that the regional balance deserves attention. Chaoyang scored the highest on the human system, indicating high levels for the population birth rate and aging rate, with certain advantages regarding the natural population growth rate and proportion of young people. Fushun achieved the lowest score for the human system, with low scores on the four indicators of birth rate, natural population growth rate, proportion of teenagers in the population, and aging rate of the population, leading to a prominent structural contradiction in the population.

The difference between the maximum and minimum scores of the residential system was 0.222, with a standard deviation of 0.071. The differences in the residential system among cities were small and the regions were relatively balanced. Shenyang achieved the highest score in the residential system due to its advantages in the four indicators of per capita living area, housing security investment, number of comprehensive service facilities in urban communities, and proportion of housing investment in GDP. Fushun achieved the lowest score, with low scores on the four indicators of population density, housing price index, per capita living space and housing security investment. These findings suggest that, in the future, it will be necessary to increase housing security investment, take effective measures to control house prices within a reasonable range, improve residents' living conditions, and expand the per capita living area.

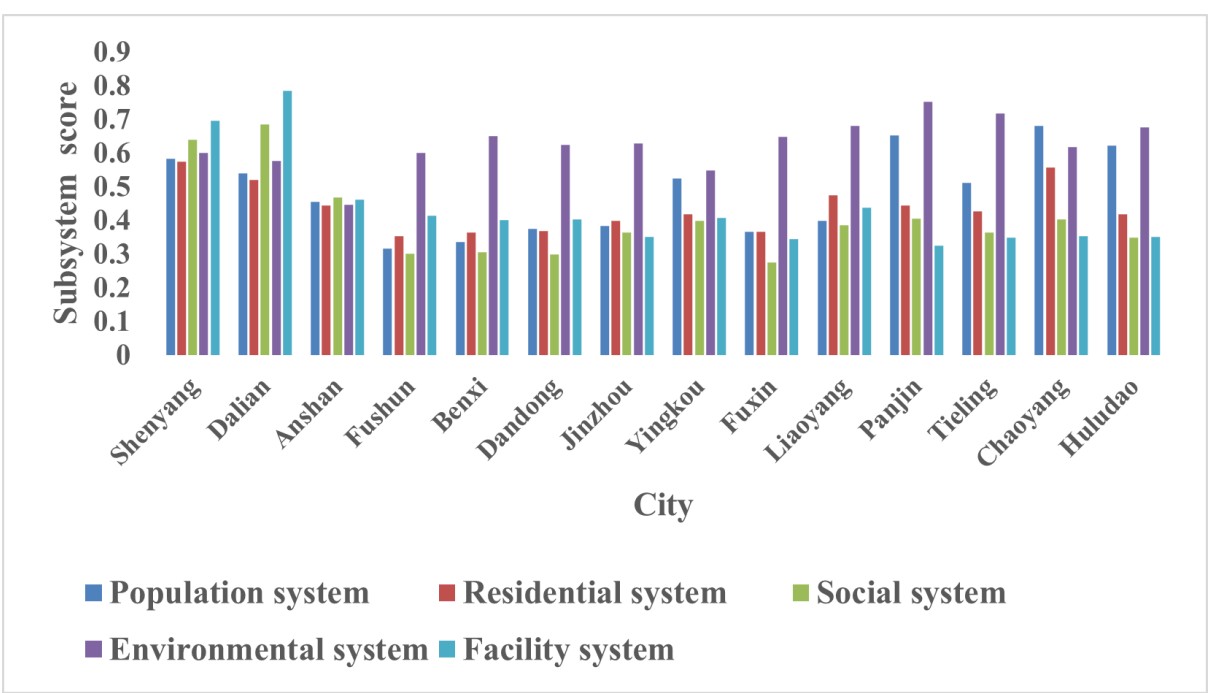

**Figure 5.** Score of each city in the subsystem.

The difference between the maximum and minimum scores of the social system was 0.409, with a standard deviation of 0.122. The differences in the social systems among the cities were significant, indicating that the regional balance deserves attention. Dalian achieved the highest score on the social system due to its advantages in unemployment rate, per capita disposable income of urban residents, medical insurance participation, proportion of scientific and technological expenditure, and other indicators. Fuxin had the lowest score, with a low score on the four indicators of per capita disposable income of urban residents, basic old-age insurance participation of urban and rural residents, medical insurance participation, and the proportion of scientific and technological expenditure.

The difference between the maximum and minimum scores in the environmental system was 0.305, with a standard deviation of 0.075. The difference in the environmental system among cities was small and the regions were relatively balanced. Panjin achieved the highest score in the environmental system due to its advantages in industrial sulfur dioxide emissions, industrial dust emissions, the centralized treatment rate of urban domestic sewage, and safe treatment of domestic garbage. During the evaluation period, its air quality also improved significantly. Anshan scored the lowest, with low scores on industrial sulfur dioxide emissions, industrial dust emissions, air quality, and other indicators. During the evaluation period, its safe treatment rate of domestic garbage dropped significantly, from first place in 2009 to thirteenth place in 2020.

The difference between the facilities systems' maximum and minimum scores was 0.458, with a standard deviation of 0.137. There were significant differences in the facilities systems between cities, indicating that the regional balance deserves attention. Dalian achieved the highest score in the social system due to its advantages regarding the level of public transportation per 10,000 people, per capita urban road area, number of hospital beds per 1000 people, health institutions, centers for disease control and prevention, internet broadband access, and public libraries per 10,000 people. In particular, Dalian consistently ranked first regarding the level of public transportation per 10,000 people and the number of sports venues during the evaluation period. Panjin achieved the lowest score, with low scores for the number of health institutions, centers for disease control and prevention, internet broadband access, stadiums and gymnasiums, as well as the number of books in public libraries per 10,000 people.

### 3.2. Factors Influencing the Health Quality of Urban Human Settlements

Previous studies have shown that the urban human settlement environment is a giant complex system affected by environmental, economic, social, political, and other factors [48–51]. Based on previous research [5], this study evaluated the quality of urban human settlements considering four dimensions: the level of urban economic development, natural background conditions, the role of government, and size of the urban population. We selected per capita GDP and regional GDP to represent the level of economic development. The average temperature represents the natural background conditions, environmental protection expenditure and general public service expenditure represent the government's role, and the year-end resident population represents the urban population size. The original value of the explanatory variables was logarithmically calculated to eliminate dimensional influence.

The $R^2$ value of the model is 0.889 and the adjusted $R^2$ value is 0.794, indicating that the model has a good fit. Table 3 shows that factors such as per capita GDP, regional GDP, environmental protection expenditure, general public service expenditure, and year-end resident population had a high positive correlation with the quality of urban human settlements. Among them, the correlation between regional GDPs was the highest. Natural background conditions had the weakest impact on the quality of human settlements.

**Table 3.** Factors influencing the quality of human settlements.

| Influencing Factors | | Pearson Correlation | Sig. |
| --- | --- | --- | --- |
| Urban economic development | Per capita GDP | 0.572 | 0.015 |
| | Regional GDP | 0.854 | 0.032 |
| Natural conditions | Average temperature | 0.425 | 0.386 |
| Role of government | Environmental protection expenditure | 0.844 | 0.035 |
| | General public service expenditure | 0.863 | 0.041 |
| Size of urban population | Rear-end resident population | 0.783 | 0.047 |

The economy is the foundation of human settlement construction. A certain level of economic strength is necessary to continuously enhance infrastructure construction, optimize the living environment, and create conditions for citizens to enjoyably pursue their spirituality. Dalian and Shenyang both had relatively high levels of economic strength, with high per capita GDP rankings and clear index advantages in social and facilities systems, as well as relatively high health levels. As one of the traditional old industrial bases in China, Liaoning has experienced a serious economic downturn in recent years, which has also led to a decline in the quality of human settlements. The construction and improvement of human settlements is a systematic project, which cannot be separated from the top-level design, policy, and macro-promotion of the government. Residents are not only the main body of the environment of a human settlement, but also the driving force for the construction of healthy settlements. To some extent, the concentration of a certain number of permanent residents illustrates the attractiveness of urban settlements to the population. Moreover, as residents' awareness of their own health improves, demand may grow for the city to continuously make its environment healthier.

## 4. Discussion

### 4.1. Construction of Index System

Previous studies have mainly constructed indicator systems based on health risks and protections, which speak to the passive construction of the environment. According to the WHO's definition of "health," "Health is not just absence of disease and physical weakness, but a state of optimal physical, mental and social well-being" [52]. Healthy human settlements can not only protect residents from adverse health factors, but also improve health by encouraging residents to practice healthy behaviors [53], which can greatly improve physical and mental health.

This study explored the quality of the five constituent systems of human settlements from three aspects: health risk, health protection, and health promotion. An evaluation index system for the quality of urban human settlements was constructed based on active intervention in space optimization. This system mainly emphasizes environmental factors that encourage individuals to practice a healthy lifestyle. It reflects urban human settlements' active intervention and guidance to improve residents' health, providing a more comprehensive evaluation of the quality and a reference for the construction of healthy human settlements.

*4.2. Limitations*

This study had a few limitations. Domestic research on the quality of urban human settlements is in its infancy, and relatively few empirical studies have been carried out. There is still room to improve the analysis and construction of the index system. Based on the spatial scale of the study and the availability and authority of data, the study mainly selected measurable factors and neglected indicators that were difficult to measure, such as nearby relationships and socio-cultural factors. There are still deficiencies in the analysis and the construction of the indicator system; we hope to complete a more detailed investigation in follow-up research.

*4.3. Recommendations*

As noted above, urban human settlements are closely related to physical and mental health. At present, urbanization continues to advance worldwide. In the rapid urbanization process, the "high pollution, high consumption" urban development model has led to the destruction of the health of the urban habitat. The "urban disease" problem is increasingly prominent, bringing great danger and serious challenges to human health. In this context, there is an urgent need to study the healthiness of urban habitats. To this end, we would like to make the following suggestions based on the findings of the current study.

(1) In future urban construction, it is necessary to consider the construction of urban community comprehensive service facilities, per capita public green space areas, and other indicators closely related to life and health, and to strive to reduce industrial dust emissions, improve air quality, and create healthy human settlements.

(2) Against the background of the Healthy China initiative and increased urbanization, cities and regions in Liaoning Province should connect and cooperate to improve the health quality of local human settlements. Specifically, core cities should lead marginal cities in building a healthy Liaoning.

(3) Cities need to take targeted measures based on their own conditions. For example, Fushun needs to take effective measures to solve its population structure contradictions by implementing a three-child policy to improve its birth rate, natural population growth rate, and proportion of teenagers, while striving to reduce its aging rate. Meanwhile, Fuxin needs to strengthen its social system, increase employment opportunities, reduce its unemployment rate, and increase the per capita disposable income of its residents. Anshan should focus on improving its environmental quality, reducing industrial sulfur dioxide and dust emissions, and improving its air quality. Additionally, Panjin must strengthen its facilities system, increase its number of health institutions and centers for disease control and prevention, meet the health needs of its residents, increase its number of stadiums and gymnasiums, increase the number of books collected in public libraries, and consistently meet the growing spiritual and cultural needs of its residents. Last, Benxi needs to increase its investment in housing security, strengthen the construction of urban community comprehensive service facilities, improve the living conditions of its residents, and expand its per capita living space.

## 5. Conclusions

To uncover the spatial-temporal and systematic differences and other factors influencing health in urban human settlements in 14 prefecture-level cities in Liaoning Province, this study explored the quality of the five constituent systems of human settlements from three aspects—health risk, health protection, and health promotion—using the entropy weight method and geographic information technology. We obtained the following findings.

(1) Regarding temporal differentiation characteristics, based on the average value over the evaluation period, the overall health quality of the urban human settlements in Liaoning Province was moderate. Based on the average value in each time period, the index of urban human settlement health in Liaoning Province clearly fluctuated and demonstrated a downward trend during a rising process of fluctuation.

(2) Regarding spatial differentiation characteristics, from 2009 to 2019, the health quality of urban human settlements in Liaoning Province showed clear regional differentiation, forming a "core-edge" pattern; that is, a spatial distribution pattern of attenuation from the core area to the edge area. Most of the urban human settlements with high health quality were in Shenyang, Dalian, and surrounding cities. The areas with cities with low levels of health quality were relatively solidified, showing a spatial distribution trend of "low in the east and high in the west" and "raised in the middle and slightly low at both ends."

(3) Regarding subsystem evolution characteristics, during the evaluation period, the health degree of urban human settlements in Liaoning Province showed clear systematic differentiation. Different cities have different advantages and disadvantages in terms of subsystems. Chaoyang scored the highest for population systems, Shenyang had the highest score for the residential system, Dalian had the highest score for the social system, Panjin scored the highest for the environmental system, and the highest score for the facility system went to Dalian.

(4) The analysis of influencing factors showed that factors such as per capita GDP, gross regional product, environmental protection expenditure, general public service expenditure, and year-end resident population had high positive correlations with the health of urban human settlements. Among them, regional GDP and environmental protection expenditures had the highest correlations. Economic development is still the precondition for improving the health of human settlements. Only by improving the economic development level of each city can we effectively improve the natural and human environment and ultimately enhance the health of human settlements.

Human settlements have an important impact on physical and mental health. In future urban construction, cities need planning strategies that include health promotion to address the health threats of rapid urbanization and low physical activity among residents. Liaoning Province is an old industrial base in China, and its city type is very representative. Through the research, this paper has drawn some basic conclusions that have practical significance and can provide reference for the improvement of the quality of human settlements in the same type of cities.

**Author Contributions:** Conceptualization, C.Z. and L.W.; methodology, C.Z.; software, C.Z.; validation, C.Z.; formal analysis, C.Z.; investigation, C.Z.; resources, C.Z.; data curation, C.Z.; writing—original draft preparation, C.Z.; writing—review and editing, C.Z.; visualization, C.Z.; supervision, L.W.; project administration, L.W. All authors have read and agreed to the published version of the manuscript.

**Funding:** This study was funded by the Foundation of Social Science Planning Foundation of Liaoning Province (L22BRK001).

**Institutional Review Board Statement:** Not applicable.

**Informed Consent Statement:** Not applicable.

**Data Availability Statement:** The data presented in this study are available on request from the corresponding author.

**Conflicts of Interest:** The authors declare no conflict of interest.

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
