# Peer review of "Evaluating the Health of Urban Human Settlements"

_sustainability, doi:10.3390/su15043042_

Round 1

Reviewer 1 Report (Previous Reviewer 2)

The authors should make the link between their theme and sustainability clearer. The 17 SDGs can be useful for this, since their theme can be directly related to some of them. Use the introduction to show this.

The methods section must detail all the steps followed to conduct the research. Use references to base the procedures performed and explain what was done in the methods section. This is the main weakness of the paper. 

Author Response

Dear editors and Reviewers:

Thank you very much for your review and comments on the manuscript. We are honored to see that you think this study has very important practical significance. All the questions you mentioned are important for us to improve the quality of the paper, and all your questions have been seriously considered and answered in detail. We hope this satisfies readers' needs for these content.

Here, we attached the revised manuscript in the editable words for your approval. A document answering every question from the referees was also summarized and enclosed. If you have any questions, please contact us without hesitation.

Comment 1

The authors should make the link between their theme and sustainability clearer. The 17 SDGs can be useful for this, since their theme can be directly related to some of them. Use the introduction to show this.

Answers to comment:

Thank you very much for editors’ reports, after consideration, We have revised the introduction section:

Original:

Urban human settlements are closely related to human health and happiness. The rapid expansion of cities and populations as a result of economic development has brought considerable challenges for human settlements, especially regarding public health [1-2]. For example, mega-scale urbanization and industrialization have triggered a series of health crises while changing the spatial forms of cities [3–6], evident in the trends of taller buildings, greater building density, and increasingly smaller living spaces, which all pose a threat to human health. According to reports, urban areas of cities have been most affected by the health challenges of the COVID-19 pandemic [7]. China’s 19th Party Congress elevated the construction of a healthy China to a national priority development strategy, emphasizing the relationship between health and the environment [8-9]. Specifically, the Healthy China 2030 initiative calls for “integrating health into the whole process of urban and rural planning, construction, and governance, and promoting the coordinated development of cities and people’s health” [10]. In light of this initiative, it is of great importance to evaluate the quality of urban human settlements in China and determine effective measures for improving their health quality.

Amendment:

Urban human settlements are closely related to human health and happiness. The rapid expansion of cities and populations as a result of economic development has brought considerable challenges for human settlements, especially regarding public health [1-2]. For example, mega-scale urbanization and industrialization have triggered a series of health crises while changing the spatial forms of cities [3–6], evident in the trends of taller buildings, greater building density, and increasingly smaller living spaces, which all pose a threat to human health. According to reports, urban areas of cities have been most affected by the health challenges of the COVID-19 pandemic [7]. The 17 sustainable development goals of the United Nations include good health and well-being, sustainable cities and communities. Healthy human settlements are inherent requirement of sustain-able cities and communities, which are conducive to promoting human health and well-being, and thus achieving sustainable development goals. China’s 19th Party Congress elevated the construction of a healthy China to a national priority development strategy, emphasizing the relationship between health and the environment [8-9]. Specifically, the Healthy China 2030 initiative calls for “integrating health into the whole process of urban and rural planning, construction, and governance, and promoting the coordinated development of cities and people’s health” [10]. In light of this initiative, it is of great importance to evaluate the quality of urban human settlements in China and determine effective measures for improving their health quality.

Comment 2

The methods section must detail all the steps followed to conduct the research. Use references to base the procedures performed and explain what was done in the methods section. This is the main weakness of the paper.

Answers to comment:

Thank you very much for editors’ reports, after consideration, we  detailed the steps followed by the research method:

Original:

Entropy method: To overcome the interference of subjective factors in determining the weight of indexes, the entropy method was used to weigh each index [41]. According to the definition of “information entropy,” the entropy value can be used to judge the dispersion degree of an indicator. The smaller the information entropy value, the greater the dispersion degree of the indicator and the greater the impact of the indicator on the comprehensive evaluation (i.e., the weight). Thus, the weighted summation method was used to measure and evaluate the degree of health in urban human settlements in Liaoning Province.

Amendment:

Entropy method: To overcome the interference of subjective factors in determining the weight of indexes, the entropy method was used to weigh each index [41]. According to the definition of “information entropy,” the entropy value can be used to judge the dispersion degree of an indicator. The smaller the information entropy value, the greater the dispersion degree of the indicator and the greater the impact of the indicator on the comprehensive evaluation (i.e., the weight). Thus, the weighted summation method was used to measure and evaluate the degree of health in urban human settlements in Liaoning Province,and is represented as follows:

(1)Standardize the data to eliminate the differences caused by different dimensions.

(2)Calculate the proportion of j indicators in the i region, m is the number of regions:

    (Equation 1)

(3)Calculate the entropy value of index j, m is the number of regions:

      (Equation 2)

     (4)Calculate the weight of index j:

     (Equation 3)

     (5)Calculate the final score of human settlements, n is the number of indicators:

    (Equation 4)

Reviewer 2 Report (New Reviewer)

In this paper, entropy method and spatial autocorrelation analysis are used to evaluate the quality of human settlements in 14 prefecture-level cities in Liaoning Province, China. The evaluation index of health quality of urban human settlement environment is constructed. It provides some ideas for the status and practice of urban human settlement environment optimization in Liaoning Province. This is a carefully done study and the findings are of considerable interest. A few minor revision are list below.

(1) What are the data obtained from the Statistical Yearbook of Liaoning Province? The source of the acquisition is vague. How to measure, the calculation needs to be detailed.

(2) In Part 3.1.1, what is the evaluation basis for the overall health quality of residential areas in Liaoning Province? Through the above reading expression is not intuitive, it is suggested to supplement and improve.

(3) In the suggestion part of 4.3, can we delete some analytical expressions and further condense the statements?

(4) The author's article still has a small part of format (such as formula 1 is not centered, Table 2 is misplaced, etc.), punctuation errors, word repetition, irrelevant Spaces, etc. Please proofread carefully according to the standard format.

(5) Many charts, but lack of concrete data support, unconvincing. FIG. 3 and FIG. 4 Can you give specific values and discuss them?

(6) By studying the evolution characteristics of the sub-system, only the data results are given at the end of the paper, which lacks certain research inspiration and scientific countermeasures, so a general view can be obtained by combining with the above.

Author Response

Dear editors and Reviewers:

Thank you very much for your review and comments on the manuscript. We are honored to see that you think this study has very important practical significance. All the questions you mentioned are important for us to improve the quality of the paper, and all your questions have been seriously considered and answered in detail. We hope this satisfies readers' needs for these content.

Here, we attached the revised manuscript in the editable words for your approval. A document answering every question from the referees was also summarized and enclosed. If you have any questions, please contact us without hesitation.

Comment 1

What are the data obtained from the Statistical Yearbook of Liaoning Province? The source of the acquisition is vague. How to measure, the calculation needs to be detailed.

Answers to comment:

Thank you very much for referees’ reports. We seriously thought about the reviewer's opinion and answered the question:

Data obtained from the Statistical Yearbook of Liaoning Province: Mortality rate, Natural population growth rate (%), Birth rate (%), Population density (persons) ,Housing price index (%),Per capita living space (m2),Housing security investment (yuan),Comprehensive service facilities in urban communities, Number of criminal cases ,Unemployment rate (%),Participation of urban and rural residents in basic old-age insurance (persons),Medical insurance participation (persons), Industrial wastewater emissions (10kt),Industrial sulfur dioxide emissions (t),Industrial dust emissions (t), Per capita disposable income of urban residents (yuan),Centralized treatment rate of urban domestic sewage (%),Air quality excellent rate (%),Harmless treatment rate of garbage (%),Per capita green space (m2),ratio in constructed areas with public green space ,Hazardous waste storage (t),Per capita domestic water consumption of residents (t), Number of hospital beds per 10,000 people, Number of health institutions, Number of centers for disease control and prevention, Number of stadiums and gymnasiums.

Data were revised and adjusted using statistical methods:

Aging rate= (Number of elderly aged 60 and above/total population) * 100%

        Proportion of urban population with the lowest social security= Number of urban population with the lowest social security/total population * 100%

Proportion of teenagers in the population (%)=Number of teenagers/total population * 100%

        Number of college students per 10,000 people (persons) = Number of college students/ total population * 10000

Proportion of housing investment in GDP (%)=Total housing investment/GDP* 100%

        Proportion of expenditure on education in GDP (%)=Expenditure on education/GDP* 100%

Proportion of scientific and technological expenditure in GDP (%)=Scientific and technological expenditure/GDP* 100%

        Number of public transportations per 10,000 people= Number of public transportations/ total population* 10000

Per capita urban road area (m2) = Actual urban road area at the end of the year/ total population

        Internet penetration rate= Number of Internet broadband access households/ Total number of households* 100%

Comment 2

In Part 3.1.1, what is the evaluation basis for the overall health quality of residential areas in Liaoning Province? Through the above reading expression is not intuitive, it is suggested to supplement and improve.

Answers to comment:

Thank you very much for referees’ reports. We seriously thought about the reviewer's opinion and answered the question:

       The evaluation of the overall health quality of residential areas in Liaoning Province is based on Equation 1-4. The entropy method (formula 1-3) is used to calculate the weight of the evaluation index of the human settlements, and the evaluation of the overall health quality of residential areas in Liaoning Province is obtained through formula 4 . We have revised this section:

Original:

3.1.1 Temporal evolution characteristics

This study’s comparative analysis of the health quality of urban human settlements in Liaoning Province between 2009 and 2020revealed the following findings.( 1) The settlements had an ordinary overall level of health quality (average value of 0.4809). Nine cities, accounting for 64.29% of the total number of cities in the province, had below-average values. Meanwhile, Dalian and Shenyang had relatively high values at 0.6312 and0.6237, respectively. These higher values were found early in the evaluation period. (2) As Figure 2 shows, based on the average value of each time period, the urban human settlement health index in Liaoning Province fluctuated, showing a “M” shaped evolution trend, accompanied by a downward trend in a rising process of fluctuation. While the index declined in some years between 2009 and 2017, it generally demonstrated an upward trend. From 2017 to 2020 the index dropped significantly from 0.493 to 0.461, a decrease of 6.5%,These findings show that urban human settlement quality in Liaoning Province did not demonstrate a positive trend during this period of social and economic development.

Amendment:

3.1.1 Temporal evolution characteristics

The entropy method (formula 1-3) is used to calculate the weight of the evaluation index of the human settlements, and the evaluation of the overall health quality of residential areas in Liaoning Province is obtained through formula 4. This study’s comparative analysis of the health quality of urban human settlements in Liaoning Province between 2009 and 2020revealed the following findings. ( 1) The settlements had an ordinary overall level of health quality (average value of 0.4809). Nine cities, accounting for 64.29% of the total number of cities in the province, had below-average values. Meanwhile, Dalian and Shenyang had relatively high values at 0.6312 and0.6237, respectively. These higher values were found early in the evaluation period. (2) As Figure 2 shows, based on the average value of each time period, the urban human settlement health index in Liaoning Province fluctuated, showing a “M” shaped evolution trend, accompanied by a downward trend in a rising process of fluctuation. While the index declined in some years between 2009 and 2017, it generally demonstrated an upward trend. From 2017 to 2020 the index dropped significantly from 0.493 to 0.461, a decrease of 6.5%,These findings show that urban human settlement quality in Liaoning Province did not demonstrate a positive trend during this period of social and economic development.

Comment 3

 In the suggestion part of 4.3, can we delete some analytical expressions and further condense the statements?

Answers to comment:

Thank you very much for editors’ reports, after consideration, we have revised the suggestion section:

Original:

4.3 Recommendations

As noted above, urban human settlements are closely related to physical and mental health. At present, urbanization continues to advance worldwide. In the rapid urbanization process, the “high pollution, high consumption” urban development model has led to the destruction of the health of the urban habitat. The “urban disease” problem is increasingly prominent, bringing great danger and serious challenges to human health. In this context, there is an urgent need to study the healthiness of urban habitats. To this end, we would like to make the following suggestions based on the findings of the current study.

(1) Human settlements have an important impact on physical and mental health. In the context of the construction of a healthy China, cities need planning strategies that include health promotion to address the health threats of rapid urbanization and low physical activity among residents. In future urban construction, it is necessary to consider the construction of urban community comprehensive service facilities, per capita public green space areas, and other indicators closely related to life and health, and to strive to reduce industrial dust emissions, improve air quality, and create healthy human settlements.

(2) Against the background of the Healthy China initiative and increased urbanization, cities and regions in Liaoning Province should connect and cooperate to improve the health quality of local human settlements. Specifically, core cities should lead marginal cities in building a healthy Liaoning.

(3) Cities need to take targeted measures based on their own conditions. For example, Fushun needs to take effective measures to solve its population structure contradictions by implementing a three-child policy to improve its birth rate, natural population growth rate, and proportion of teenagers, while striving to reduce its aging rate. Meanwhile, Fuxin needs to strengthen its social system, increase employment opportunities, reduce its un-employment rate, and increase the per capita disposable income of its residents. Anshan should focus on improving its environmental quality, reducing industrial sulfur dioxide and dust emissions, and improving its air quality. Additionally, Panjin must strengthen its facilities system, increase its number of health institutions and centers for disease control and prevention, meet the health needs of its residents, increase its number of stadiums and gymnasiums, increase the number of books collected in public libraries, and consistently meet the growing spiritual and cultural needs of its residents. Last, Benxi needs to increase its investment in housing security, strengthen the construction of urban community comprehensive service facilities, improve the living conditions of its residents, and expand its per capita living space. Liaoning Province is an old industrial base in China, and its city type is very representative. Through the research, this paper has drawn some basic conclusions that have practical significance and can provide reference for the improvement of the quality of human settlements in the same type of cities. However, whether the research results have commonalities or other differences in other regions needs to be further verified by expanding the research scope. It should be pointed out that since the World Health Group put forward the concept of a healthy city, it has been widely recognized. More and more cities have begun to focus on the health of human settlements and carried out a healthy city campaign, these cities are inspiring. In Europe and the United States, since the 1980s, the healthy city movement has achieved great success in promoting health and sustainable development with the help of health promotion, urban planning and ecosystem perspectives. The European Healthy Cities Project is a social movement that promotes and maintains the health of urban populations through the use of a wide range of political, social and behavioral interventions, and emphasizes health equity [54] in all policies California builds community gardens to enhance the nutrition and physical activity of its residents[55]. Working through a healthy environment, such as the Healthy Municipalities, Cities and Communities Strategy, is one of the more successful strategies for Latin American countries to implement health promotion[56]. These methods and ideas are very enlightening and can supplement the research conclusions of this study.

Amendment:

4.3 Recommendations

As noted above, urban human settlements are closely related to physical and mental health. At present, urbanization continues to advance worldwide. In the rapid urbanization process, the “high pollution, high consumption” urban development model has led to the destruction of the health of the urban habitat. The “urban disease” problem is increasingly prominent, bringing great danger and serious challenges to human health. In this context, there is an urgent need to study the healthiness of urban habitats. To this end, we would like to make the following suggestions based on the findings of the current study.

(1) In future urban construction, it is necessary to consider the construction of urban community comprehensive service facilities, per capita public green space areas, and other indicators closely related to life and health, and to strive to reduce industrial dust emissions, improve air quality, and create healthy human settlements.

(2) Against the background of the Healthy China initiative and increased urbanization, cities and regions in Liaoning Province should connect and cooperate to improve the health quality of local human settlements. Specifically, core cities should lead marginal cities in building a healthy Liaoning.

(3) Cities need to take targeted measures based on their own conditions. For example, Fushun needs to take effective measures to solve its population structure contradictions by implementing a three-child policy to improve its birth rate, natural population growth rate, and proportion of teenagers, while striving to reduce its aging rate. Meanwhile, Fuxin needs to strengthen its social system, increase employment opportunities, reduce its un-employment rate, and increase the per capita disposable income of its residents. Anshan should focus on improving its environmental quality, reducing industrial sulfur dioxide and dust emissions, and improving its air quality. Additionally, Panjin must strengthen its facilities system, increase its number of health institutions and centers for disease control and prevention, meet the health needs of its residents, increase its number of stadiums and gymnasiums, increase the number of books collected in public libraries, and consistently meet the growing spiritual and cultural needs of its residents. Last, Benxi needs to increase its investment in housing security, strengthen the construction of urban community comprehensive service facilities, improve the living conditions of its residents, and expand its per capita living space.

Comment 4

The author's article still has a small part of format (such as formula 1 is not centered, Table 2 is misplaced, etc.), punctuation errors, word repetition, irrelevant Spaces, etc. Please proofread carefully according to the standard format.

Answers to comment:

Thank you very much for editors’ reports, we proofread carefully according to the standard format.

        Due to too many modifications, please see the attachment.

Comment 5

Many charts, but lack of concrete data support, unconvincing. FIG. 3 and FIG. 4 Can you give specific values and discuss them?

Answers to comment:

Thank you very much for editors’ reports, we seriously thought about the reviewer's opinion and answered the question.

The specific values of FIG. 3 are as follows:

2009

2011

2013

2015

2017

2019

2020

Shenyang

0.648636

0.605876

0.662924

0.596052

0.635046

0.596768

0.620477

Dalian

0.629105

0.669081

0.625825

0.614668

0.64448

0.644884

0.590161

Anshan

0.442979

0.463025

0.450582

0.457741

0.496026

0.451481

0.431247

Fushun

0.399303

0.433151

0.460369

0.441164

0.369772

0.356439

0.371643

Benxi

0.421787

0.443745

0.434125

0.392646

0.407633

0.393994

0.434528

Dandong

0.375717

0.461167

0.420493

0.428657

0.461222

0.426089

0.381425

Jinzhou

0.421992

0.464634

0.392943

0.441212

0.409195

0.473948

0.396644

Yingkou

0.482037

0.468462

0.498458

0.413676

0.508745

0.435282

0.432693

Fuxin

0.385148

0.391969

0.416273

0.420988

0.44293

0.375627

0.412301

Liaoyang

0.455443

0.461214

0.494188

0.482636

0.504444

0.49489

0.456491

Panjin

0.539834

0.524217

0.528447

0.498052

0.544926

0.524486

0.468519

Tieling

0.504278

0.50006

0.514093

0.487228

0.444015

0.454844

0.434483

Chaoyang

0.481953

0.461319

0.479194

0.591865

0.525379

0.545728

0.538129

Huludao

0.494531

0.458333

0.494831

0.489148

0.504007

0.485073

0.484869

The specific values of FIG. 4 are as follows:

Population system

Residential system

Social system

Environmental system

Facility system

2009

0.523058942

0.39404812

0.420126055

0.616255941

0.403595218

2011

0.506080324

0.46612959

0.382214277

0.613359706

0.44168063

2013

0.463390012

0.421745828

0.436816437

0.660330501

0.442076127

2015

0.448917924

0.445946382

0.426820995

0.622496906

0.447686803

2017

0.492073982

0.493297162

0.387290465

0.597816705

0.477584803

2019

0.473877928

0.445732231

0.374157789

0.651323587

0.411713794

2020

0.459432

0.392305752

0.389055105

0.620268932

0.41046055

Comment 6

By studying the evolution characteristics of the sub-system, only the data results are given at the end of the paper, which lacks certain research inspiration and scientific countermeasures, so a general view can be obtained by combining with the above

Answers to comment:

Thank you very much for editors’ reports, we seriously thought about the reviewer's opinion and answered the question. We have carefully analyzed the evolution characteristics of subsystems and modified this part.

Original:

Systematic temporal differentiation: (1) System time differentiation. As shown in Figure 5. During the study period, there were certain differences in the change trends of the five major systems of urban human settlements in Liaoning Province. (1) The overall environmental system and facility system showed a trend of fluctuation and increase, indicating that the urban environment and infrastructure construction in Liaoning Province had improved overall and improved to a certain extent from 2009 to 2020. (2) The human system, social system and residential systems have declined to a certain extent. The de-cline of the residential system was small. The downward trend of the human system was obvious, from 0.523 in 2009 to 0.459 in 2019. The main reason is that the natural population growth rate and birth rate have declined significantly, and the aging rate has continued to rise. During the evaluation period, the degree of aging in 14 prefecture level cities has continued to increase, and the trend of population structure change deserves attention. The social system declined from 0.420 in 2009 to 0.389 in 2020, mainly due to the impact of downward pressure on the economy, and some cities' scores on indicators such as the unemployment rate and the proportion of education expenditure decreased to a certain ex-tent.

Amendment:

Systematic temporal differentiation: (1) System time differentiation. As shown in Figure 5. During the study period, there were certain differences in the change trends of the five major systems of urban human settlements in Liaoning Province. (1)The overall environmental system and facility system showed a trend of fluctuation and increase, indicating that the urban environment and infrastructure construction in Liaoning Province had improved overall and improved to a certain extent from 2009 to 2020. (2) The human system, social system and residential systems have declined to a certain extent. The de-cline of the residential system was small. The downward trend of the human system was obvious, from 0.523 in 2009 to 0.459 in 2019. The main reason is that the natural population growth rate and birth rate have declined significantly, and the aging rate has continued to rise. During the evaluation period, the degree of aging in 14 prefecture level cities has continued to increase, and the trend of population structure change deserves attention. The social system declined from 0.420 in 2009 to 0.389 in 2020, mainly due to the impact of downward pressure on the economy, and some cities' scores on indicators such as the unemployment rate and the proportion of education expenditure decreased to a certain ex-tent.

The human settlement environment is composed of five systems. Only the coordinated development of the five systems can realize the continuous improvement of the quality of human settlement environment. The evaluation found that the quality of the human system and social system has declined significantly. It is necessary to optimize the fertility policy and improve the talent policy to cope with the declining proportion of the workforce. Take effective measures, such as reforming state-owned enterprises and promoting urban economic development.

Reviewer 3 Report (New Reviewer)

Dear Authors,

In accordance with the proofreading criteria of the publisher, I prepared a  report, which would be as follows:

In my opinion the content of the proposed paper meets the objectives set out in the special issue information letter.

Using the scientific methods applied in accordance with the author’s scientific objectives resulted useful scientific achievements.

The main strength of article is that the authors propose an index to evaluate the health quality of urban human settlements, with the help of which we can obtain useful information on the issue of good implementation practice for optimizing of urban human settlements.

The references used in the main chapters are relevant and assist the reader to understand the authors proposals. The illustrations used are regular.

In addition to acknowledging the high-quality work, I recommend adding a few paragraph to the chapter named as “conclusions”, in which the authors should formulate some general proposals for future research related to author’s findings. An important question is to clarify at the end of the paper, to what extent the case study of Liaoning Province helps to increase the national and international scientific potential related to the examined topic.

 Based on the above, I suggest to publish the reviewed article.

Author Response

Dear editors and Reviewers:

Thank you very much for your review and comments on the manuscript. We are honored to see that you think this study has very important practical significance. All the questions you mentioned are important for us to improve the quality of the paper, and all your questions have been seriously considered and answered in detail. We hope this satisfies readers' needs for these content.

Here, we attached the revised manuscript in the editable words for your approval. A document answering every question from the referees was also summarized and enclosed. If you have any questions, please contact us without hesitation.

In addition to acknowledging the high-quality work, I recommend adding a few paragraph to the chapter named as “conclusions”, in which the authors should formulate some general proposals for future research related to author’s findings. An important question is to clarify at the end of the paper, to what extent the case study of Liaoning Province helps to increase the national and international scientific potential related to the examined topic.

Answers to comment:

        Thank you very much for editors’ reports, we seriously thought about the reviewer's opinion and answered the question.

        In the conclusion, we add the following paragraph:

Human settlements have an important impact on physical and mental health. In future urban construction, cities need planning strategies that include health promotion to address the health threats of rapid urbanization and low physical activity among residents. Liaoning Province is an old industrial base in China, and its city type is very representative. Through the research, this paper has drawn some basic conclusions that have practical significance and can provide reference for the improvement of the quality of hu-man settlements in the same type of cities.

Round 2

Reviewer 1 Report (Previous Reviewer 2)

The authors addressed the comments presented. 

This manuscript is a resubmission of an earlier submission. The following is a list of the peer review reports and author responses from that submission.

Round 1

Reviewer 1 Report

Review Report on “Evaluating the health of urban human settlements by

Zhang and Wang-2004159”

In this study, the authors aimed to propose effective measures to improve the quality of urban human settlements and to provide theoretical support for the healthy development of such settlements.

The methodologies used to perform the work are appropriate. The study and results are regional in scope but unique in the literature. The paper presents new information. Besides, the paper is structurally very well designed.

Therefore, the presented paper by Zhang and Wang is worthy of publication with this shape in Sustainability Journal.

Author Response

Thank you for your positive review of the study. In your evaluation, the score of references is relatively low.After careful reading and sorting of the journal literature, we added 16 references and the references have been reviewed and the bibliography has been updated. Thanks again for your recognition.

References:

1.Avila-Palencia I, Sánchez B N, Rodríguez D A, et al. Health and Environmental Co-Benefits of City Urban Form in Latin America: An Ecological Study [J]. Sustainability 2022, 14(22), 14715; https://doi.org/10.3390/su142214715

2.Blanco E, Pedersen Zari M, Raskin K, et al. Urban ecosystem-level biomimicry and regenerative design: Linking ecosystem functioning and urban built environments[J]. Sustainability, 2021, 13(1): 404.

8.Alberti M. The effects of urban patterns on ecosystem function[J]. International regional science review, 2005, 28(2): 168-192.

9.Blanco E, Pedersen Zari M, Raskin K, et al. Urban ecosystem-level biomimicry and regenerative design: Linking ecosystem functioning and urban built environments[J]. Sustainability, 2021, 13(1): 404.

31.Landrigan P J. Air pollution and health[J]. The Lancet Public Health, 2017, 2(1): e4-e5.

32.O'Neill M S, Jerrett M, Kawachi I, et al. Health, wealth, and air pollution: advancing theory and methods[J]. Environmental health perspectives, 2003, 111(16): 1861-1870.

33.Sarkodie S A, Owusu P A. Global assessment of environment, health and economic impact of the novel coronavirus (COVID-19)[J]. Environment, Development and Sustainability, 2021, 23(4): 5005-5015.

34.Emanuel E J, Persad G, Upshur R, et al. Fair allocation of scarce medical resources in the time of Covid-19[J]. New England Journal of Medicine, 2020, 382(21): 2049-2055.

35.Krütli P, Rosemann T, Törnblom K Y, et al. How to fairly allocate scarce medical resources: ethical argumentation under scrutiny by health professionals and lay people[J]. PloS one, 2016, 11(7): e0159086.

36.Ma Y, Liang H, Li H, et al. Towards the healthy community: Residents’ perceptions of integrating urban agriculture into the old community micro-transformation in Guangzhou, China[J]. Sustainability, 2020, 12(20): 8324.

40.Zhao J, Ji G, Tian Y, et al. Environmental vulnerability assessment for mainland China based on entropy method[J]. Ecological Indicators, 2018, 91: 410-422.

45.Olive D J. Multiple linear regression[M]//Linear regression. Springer, Cham, 2017: 17-83.

46.Grégoire G. Multiple linear regression[J]. European Astronomical Society Publications Series, 2014, 66: 45-72.

47.Sari A I, Suwarto S, Suminah S, et al. Empowering the Community in the Use of Livestock Waste Biogas as a Sustainable Energy Source[J]. Sustainability, 2022, 14(21): 14121.

51.Leonardi F. The definition of health: towards new perspectives[J]. International Journal of Health Services, 2018, 48(4): 735-748.

52.Kim E S, Kubzansky L D, Soo J, et al. Maintaining healthy behavior: a prospective study of psychological well-being and physical activity[J]. Annals of Behavioral Medicine, 2017, 51(3): 337-347.

Reviewer 2 Report

The theme of the paper is interesting; however, my main concern is regarding the contributions of the findings to the literature. The analysis provided is too simple to justify the publication. Maybe authors could perform a survey or a Delphi study with experts in the area to develop a route to be followed. Or compare the situation of different countries.

Why the analysis is up to 2019 and not up to 2021 or 2022? Authors must update the data or justify the reason for using this period in the text. Indeed, the authors even mention the covid-19 pandemic that became a worldwide problem in 2020.

The methods must be better explained. There are texts without references and the authors do not justify the procedures selected to perform the research.

All the steps followed to perform the research must be detailed in the methods. The current tent does not enable the research replication.

The results presentation, implications and contributions of the research also need to be improved.

Author Response

Comment 1

The theme of the paper is interesting; however, my main concern is regarding the contributions of the findings to the literature. The analysis provided is too simple to justify the publication. Maybe authors could perform a survey or a Delphi study with experts in the area to develop a route to be followed. Or compare the situation of different countries.

Answers to comment 1:

Thank you very much for referees’ reports. We seriously thought about the reviewer's opinion and answered the question. Because of the epidemic, many cities have adopted strict prevention and control policies, including the cities in the study area. Under such conditions, it is very difficult for us to carry out the survey, which is a pity in the research. Regarding to ‘compare the situation of different countries’, due to the differences in statistical caliber, it is difficult to make comparisons between countries.  The comments of reviewers are very valuable and enlightening to me. Based on the comment, we have made the following adjustments to the Limitations:

Original:

LIMITATIONS:

This study has certain limitations. Domestic research on the quality of urban human settlements is in its infancy, and there are relatively few empirical studies. There is still room for improvement in the construction of the index system. Due to data availability, the residents’ health factors were not considered. Future research should collect innovative quantitative index data from more angles to improve the research results so that the research conclusions can better serve the government’s policy formulation.

Amendment:

LIMITATIONS:

This study had a few certain limitations. Domestic research on the quality of urban human settlements is in its infancy, and relatively few empirical studies have been done. There is still room to improve the analysis and construction of the index system. Based on the spatial scale of the study and the availability and authority of data, the study mainly selected measurable factors and neglected indicators that were difficult to measure, such as nearby relationships and socio-cultural factors. Due to limitations related to paper length and the research focus, this study only conducted a spatial-temporal evolution analysis of the factors influencing the levels of health in urban human settlements in Liaoning Province. There are still deficiencies in the analysis and the construction of the indicator system; we hope to complete a more detailed investigation in follow-up research. Further work should also be done to improve the analysis and indicator system through a survey, a Delphi study, or other methods.

Comment 2

Why the analysis is up to 2019 and not up to 2021 or 2022? Authors must update the data or justify the reason for using this period in the text. Indeed, the authors even mention the covid-19 pandemic that became a worldwide problem in 2020.

Answers to comment 2:

Thank you very much for referees’ reports. We seriously thought about the reviewer's opinion and answered the question. The impact of urban human settlements on residents' health has a lagging effect, which is mainly manifested in the form that after the change of urban human settlements, it will not have an immediate effect on residents' health, and its impact will not disappear in a short period of time. In the next few years, it will also have a significant effect on residents' health. the covid-19 pandemic that became a worldwide problem in 2020, therefore, it is necessary to evaluate the health of the human settlements before the epidemic.

After careful consideration, we think it is necessary to state the study period in the Evaluation system and Data sources.

Original:

EVALUATION SYSTEM AND DATA SOURCES.

The index data were derived from the statistical yearbook of the Liaoning province from 2010 to 2020. Due to differences in the quality of the statistics, some data were revised and adjusted using statistical methods, referring to the Statistical Bulletin of the National Economic and Social Development of each city. Specifically, missing data were filled with the average growth rates since 2005. The data acquisition in this study followed the principles of authority and reliability, which can truly reflect the quality of urban hu-man settlements in the Liaoning province.

Amendment:

EVALUATION SYSTEM AND DATA SOURCES.

The index data were derived from the statistical yearbooks for the Liaoning Province from 2010 to 2020. Due to differences in the quality of the statistics, some data were revised and adjusted using statistical methods, referring to the Statistical Bulletin of the National Economic and Social Development of each city. Specifically, missing data were filled with the average growth rates since 2005. The data acquisition in this study followed the principles of authority and reliability, which can truly reflect the quality of urban human settlements in Liaoning Province.

It is important to note that urban human settlements have a lagged effect on residents' health, which is mainly evident in that changes in urban human settlements do not immediately affect residents' health, but do eventually have an impact does not quickly disappear. Given the emergence of the COVID-19 pandemic in 2020, which sharply impacted human health, it is necessary to evaluate the health of human settlements before the pandemic.

Comment 3

The methods must be better explained. There are texts without references and the authors do not justify the procedures selected to perform the research

All the steps followed to perform the research must be detailed in the methods. The current tent does not enable the research replication..

Answers to comment 3

Thank you very much for referees’ reports, we seriously thought about the reviewer's opinion and revised the METHODS:

Original:

METHODS:

Entropy method: To overcome the interference of subjective factors in determining the weight of indexes, the entropy method was used to weigh each index. Based on this, the weighted summation method was used to measure and evaluate the health degree of urban human settlements in the Liaoning province.

Multiple linear regression: In real society, the emergence of a phenomenon is often not the result of a single factor, but rather a result of the joint action of multiple factors. Human settlement environment is a complex giant system, and its influencing factors are also characterized by pluralism and complexity. As such, changes in dependent variables must be reflected through multiple independent variables. Therefore, this study uses multiple linear regression to study the factors that affect the differences in human settlement environment quality in Liaoning’s cities.

Amendment:

METHODS:

Entropy method: To overcome the interference of subjective factors in determining the weight of indexes, the entropy method was used to weigh each index[40]. According to the definition of “information entropy,” the entropy value can be used to judge the dispersion degree of an indicator. The smaller the information entropy value, the greater the dispersion degree of the indicator and the greater the impact of the indicator on the com-prehensive evaluation (i.e., the weight). Thus, the weighted summation method was used to measure and evaluate the degree of health in urban human settlements in Liaoning Province.

References:

Zhao J, Ji G, Tian Y, et al. Environmental vulnerability assessment for mainland China based on entropy method[J]. Ecological Indicators, 2018, 91: 410-422.

Multiple linear regression: In a real society, the emergence of a phenomenon is often not the result of a single factor, but rather a result of the joint action of multiple factors. A human settlement environment is a giant complex system, and its influencing factors are also characterized by pluralism and complexity[45-47]. As such, changes in dependent variables must be reflected through multiple independent variables. Therefore, this study used multiple linear regression to study the factors affecting the differences in the environmental quality of the human settlements across Liaoning Province’s cities.

References:

45.Olive D J. Multiple linear regression[M]//Linear regression. Springer, Cham, 2017: 17-83.

46.Grégoire G. Multiple linear regression[J]. European Astronomical Society Publications Series, 2014, 66: 45-72.

47.Sari A I, Suwarto S, Suminah S, et al. Empowering the Community in the Use of Livestock Waste Biogas as a Sustainable Energy Source[J]. Sustainability, 2022, 14(21): 14121.

Comment 4

The results presentation, implications and contributions of the research also need to be improved.

Answers to comment 4

Thank you very much for editors’ reports, After consideration, we have resorted the discussion and conclusion part and made the following modifications

Original:

DISCUSSION

Construction of index system

Previous studies have mainly constructed indicator systems from the aspects of health risk and protection, which encapsulate the idea of passive environment construction. This study explored the quality of the five constituent systems of human settlements from three aspects: health risk, health protection, and health promotion. An evaluation index system for the quality of urban human settlements was constructed based on active intervention in space optimization. This system mainly emphasizes environmental fac-tors that encourage individuals to practice a healthy lifestyle. It reflects urban human settlements’ active intervention and guidance to improve residents’ health, providing a more comprehensive evaluation of the quality and a reference for the construction of healthy human settlements.

Limitations

This study has certain limitations. Domestic research on the quality of urban human settlements is in its infancy, and there are relatively few empirical studies. There is still room for improvement in the construction of the index system. Due to data availability, the residents’ health factors were not considered. Future research should collect innovative quantitative index data from more angles to improve the research results so that the research conclusions can better serve the government’s policy formulation.

Recommendations

Urban human settlements are closely related to physical and mental health. At pre-sent, urbanization continues to advance worldwide. In the rapid urbanization process, the “high pollution, high consumption” urban development model has led to the destruction of the health of the urban habitat. The “urban disease” problem is increasingly prominent, bringing great danger and serious challenges to human health. In this context, there is an urgent need to study the healthiness of urban habitats. To this end, the present study suggests:

(1) In future urban construction, it is necessary to consider the construction of urban community comprehensive service facilities, per capita public green space areas, and other indicators closely related to people’s lives and health, and to strive to reduce industrial dust emissions, improve air quality, and create healthy human settlements.

(2) Against the background of the Healthy China 2030 initiative and increased urbanization, cities and regions in the Liaoning province should enhance connections and cooperation to improve the health degree of human settlements in marginal cities through the leading role of core cities to build a healthy Liaoning.

(3) Cities need to take targeted measures based on their own conditions. For example, Fushun needs to take effective measures to solve the population structure contradictions by implementing the three-child policy to improve the birth rate, natural population growth rate, and proportion of teenagers in the population, while striving to reduce the aging rate. Fuxin needs to strengthen the construction of the social system, increase employment opportunities, reduce the unemployment rate, and increase the per capita dis-posable income of residents. Anshan should focus on improving its environmental quality, reducing industrial sulfur dioxide and dust emissions, and improving air quality. Panjin must strengthen the construction of the facilities system, increase the number of health institutions and centers for disease control and prevention, meet the health needs of residents, strengthen the construction of the number of stadiums and gymnasiums, increase the number of books collected in public libraries, and consistently meet the growing spiritual and cultural needs of its residents. Benxi needs to increase investment in housing security, strengthen the construction of urban community comprehensive service facilities, improve the living conditions of its residents, and expand the per capita living space.

Original:

CONCLUSIONS

Using the entropy weight method and geographic information technology for 14 prefecture-level cities in the Liaoning province, this study empirically investigated spatialtemporal and systematic differences and identified factors influencing the quality levels of urban human settlements based on a health perspective. We obtained the following findings:

(1) Regarding temporal differentiation characteristics, based on the average value over the evaluation period, the health degree of urban human settlements in the Liaoning province was moderate. From the average value of each time period, the index of urban human settlements health in the Liaoning province fluctuated obviously, with a certain downward trend in the rising process of fluctuation.

(2) Regarding spatial differentiation characteristics, from 2009 to 2019, the health degree of urban human settlements in the Liaoning province showed obvious regional differentiation, forming a “core-edge” pattern, that is, a spatial distribution pattern of attenuation from the core area to the edge area. Most high-value areas of urban human settlements health were concentrated in Shenyang, Dalian, and surrounding cities. The cities in the low-value areas were relatively solidified, showing a spatial distribution trend of “low in the east and high in the west” and “raised in the middle and slightly low at both ends.”

(3) Regarding subsystem evolution characteristics, during the evaluation period, the health degree of urban human settlements in the Liaoning province showed obvious systematic differentiation. Chaoyang scored the highest for population systems; Shenyang had the highest score in the residential system; Dalian had the highest score for the social system; Panjin scored the highest for the environmental system; and the highest score for the facility system went to Dalian.

(4) The analysis of influencing factors showed that factors such as per capita GDP, gross regional product, environmental protection expenditure, general public service expenditure, year-end resident population, and other factors had high positive correlations with the health of urban human settlements. Among them, the regional GDP and environmental protection expenditures had the highest correlations.

Amendment:

DISCUSSION

Construction of index system

Previous studies have mainly constructed indicator systems from the aspects based on health risks and protections, which speak to the idea of the passive environment construction of the environment. According to the WHO’s definition of “health,” “Health is not just absence of disease and physical weakness, but a state of optimal physical, mental and social well-being” [51]. Healthy human settlements can not only protect residents from adverse health factors, but also improve health by encouraging residents to practice healthy behaviors [52] , which can greatly improve physical and mental health. This study explored the quality of the five constituent systems of human settlements from three aspects: health risk, health protection, and health promotion. An evaluation index system for the quality of urban human settlements was constructed based on active intervention in space optimization. This system mainly emphasizes environmental factors that encourage individuals to practice a healthy lifestyle. It reflects urban human settlements’ active intervention and guidance to improve residents’ health, providing a more comprehensive evaluation of the quality and a reference for the construction of healthy human settlements.

References:

  1. Leonardi, F. The definition of health: Towards new perspectives. Int. J. Health Serv. 2018, 48, 735–748.
  2. Kim, E.S.; Kubzansky, L.D.; Soo, J.; Boehm, J.K. Maintaining healthy behavior: A prospective study of psychological well-being and physical activity. Ann. Behav. Med. 2017, 51, 337–347.

Limitations

This study had a few certain limitations. Domestic research on the quality of urban human settlements is in its infancy, and relatively few empirical studies have been done. There is still room to improve the analysis and construction of the index system. Based on the spatial scale of the study and the availability and authority of data, the study mainly selected measurable factors and neglected indicators that were difficult to measure, such as nearby relationships and socio-cultural factors. Due to limitations related to paper length and the research focus, this study only conducted a spatial-temporal evolution analysis of the factors influencing the levels of health in urban human settlements in Liaoning Province. There are still deficiencies in the analysis and the construction of the indicator system; we hope to complete a more detailed investigation in follow-up research. Further work should also be done to improve the analysis and indicator system through a survey, a Delphi study, or other methods.

Recommendations

As noted above, urban human settlements are closely related to physical and mental health. At present, urbanization continues to advance worldwide. In the rapid urbaniza-tion process, the “high pollution, high consumption” urban development model has led to the destruction of the health of the urban habitat. The “urban disease” problem is in-creasingly prominent, bringing great danger and serious challenges to human health. In this context, there is an urgent need to study the healthiness of urban habitats. To this end, we would like to make the following suggestions based on the findings of the current study.

(1) Human settlements have an important impact on physical and mental health. In the context of the construction of a healthy China, cities need planning strategies that in-clude health promotion to addresses the health threats of rapid urbanization and low physical activity among residents.

(2) In future urban construction, it is necessary to consider the construction of urban community comprehensive service facilities, per capita public green space areas, and oth-er indicators closely related to life and health, and to strive to reduce industrial dust emis-sions, improve air quality, and create healthy human settlements.

(3) Against the background of the Healthy China initiative and increased urbaniza-tion, cities and regions in Liaoning Province should connect and cooperate to improve the health quality of local human settlements. Specifically, core cities should lead marginal cities in building a healthy Liaoning.

(4) Cities need to take targeted measures based on their own conditions. For example, Fushun needs to take effective measures to solve its population structure contradictions by implementing a three-child policy to improve its birth rate, natural population growth rate, and proportion of teenagers, while striving to reduce its aging rate. Meanwhile, Fuxin needs to strengthen its social system, increase employment opportunities, reduce its un-employment rate, and increase the per capita disposable income of its residents. Anshan should focus on improving its environmental quality, reducing industrial sulfur dioxide and dust emissions, and improving its air quality. Additionally, Panjin must strengthen its facilities system, increase its number of health institutions and centers for disease con-trol and prevention, meet the health needs of its residents, increase its number of stadiums and gymnasiums, increase the number of books collected in public libraries, and consist-ently meet the growing spiritual and cultural needs of its residents. Last, Benxi needs to increase its investment in housing security, strengthen the construction of urban commu-nity comprehensive service facilities, improve the living conditions of its residents, and expand its per capita living space.

CONCOUSIONS

To uncover the spatial-temporal and systematic differences and other factors influencing health in urban human settlements in 14 prefecture-level cities in Liaoning Province, this study explored the quality of the five constituent systems of human settlements from three aspects—health risk, health protection, and health promotion—using the entropy weight method and geographic information technology. We obtained the following findings.

(1) Regarding temporal differentiation characteristics, based on the average value over the evaluation period, the overall health quality of the urban human settlements in Liaoning Province was moderate. Based on the average value in each time period, the index of urban human settlement health in Liaoning Province obviously fluctuated and demonstrated a downward trend during a rising process of fluctuation.

(2) Regarding spatial differentiation characteristics, from 2009 to 2019, the health quality of urban human settlements in Liaoning Province showed obvious regional differentiation, forming a “core-edge” pattern; that is, a spatial distribution pattern of attenuation from the core area to the edge area. Most of the urban human settlements with high health quality were in Shenyang, Dalian, and surrounding cities. The areas with cities with low levels of health quality were relatively solidified, showing a spatial distribution trend of “low in the east and high in the west” and “raised in the middle and slightly low at both ends.”

(3) Regarding subsystem evolution characteristics, during the evaluation period, the health degree of urban human settlements in Liaoning Province showed obvious systematic differentiation. Different cities have different advantages and disadvantages in terms of subsystems. Chaoyang scored the highest for population systems, Shenyang had the highest score for the residential system, Dalian had the highest score for the social system, Panjin scored the highest for the environmental system, and the highest score for the facility system went to Dalian.

(4) The analysis of influencing factors showed that factors such as per capita GDP, gross regional product, environmental protection expenditure, general public service expenditure, and year-end resident population had high positive correlations with the health of urban human settlements. Among them, regional GDP and environmental protection expenditures had the highest correlations. Economic development is still the pre-condition for improving the health of human settlements. Only by improving the eco-nomic development level of each city can we effectively improve the natural and human environment and ultimately enhance the health of human settlements.

Comment 4

Moderate English changes required

Thank you very much for editors’ reports. To improve the quality of language expression, we used the editing services to check the manuscript.

Reviewer 3 Report

In the paper the idea of urban health is limited to the quantitatively measurable facts. Certainly this is one quality of the paper... But there are  invisible factors and non-measurable environmental features, e.g. walkability, beauty (of architecture and urban design, etc.), green neighborhoods, neighborhood relations and socio-cultural factors, etc. which in fact influence the overall quality of life in cities and settlements which in turn constitute the basis for personal health and happiness of the citizens.

Author Response

Comment

In the paper the idea of urban health is limited to the quantitatively measurable facts. Certainly this is one quality of the paper... But there are  invisible factors and non-measurable environmental features, e.g. walkability, beauty (of architecture and urban design, etc.), green neighborhoods, neighborhood relations and socio-cultural factors, etc. which in fact influence the overall quality of life in cities and settlements which in turn constitute the basis for personal health and happiness of the citizens.

Answers to comment:

Thank you very much for referees’ reports. We seriously thought about the reviewer's opinion and answered the question. Based on the spatial scale of the study and the availability and authority of data, the paper mainly selects measurable factors. But as the reviewers said, walkability, beauty (of architecture and urban design, etc.), green neighborhoods, neighborhood relations and socio-cultural factors, etc. which in fact influence the overall quality of life in cities and settlements which in turn constitute the basis for personal health and happiness of the citizens. The comments of reviewers are very valuable and enlightening to me. Based on the comment, we have made the following adjustments to the original manuscript. The limitation of this paper is improved and supplemented:

Original:

LIMITATIONS:

This study has certain limitations. Domestic research on the quality of urban human settlements is in its infancy, and there are relatively few empirical studies. There is still room for improvement in the construction of the index system. Due to data availability, the residents’ health factors were not considered. Future research should collect innovative quantitative index data from more angles to improve the research results so that the research conclusions can better serve the government’s policy formulation.

Amendment:

LIMITATIONS:

This study had a few certain limitations. Domestic research on the quality of urban human settlements is in its infancy, and relatively few empirical studies have been done. There is still room to improve the analysis and construction of the index system. Based on the spatial scale of the study and the availability and authority of data, the study mainly selected measurable factors and neglected indicators that were difficult to measure, such as nearby relationships and socio-cultural factors. Due to limitations related to paper length and the research focus, this study only conducted a spatial-temporal evolution analysis of the factors influencing the levels of health in urban human settlements in Liaoning Province. There are still deficiencies in the analysis and the construction of the indicator system; we hope to complete a more detailed investigation in follow-up research. Further work should also be done to improve the analysis and indicator system through a survey, a Delphi study, or other methods.

Round 2

Reviewer 2 Report

The authors made some improvements in the paper, however, the main weakness of the study remains. 

The two first comments I made in revision 1 were the main concern about the paper, regarding the lack of a valuable contribution that the research provide.   The authors did not provide any changes to improve these two items.   They recognized as limitations of the study. 

In this sense, the comment I sent in this round 2 ("The authors made some improvements in the paper, however, the main weakness of the study remains") was basically because the authors did not address the previous comments.   The study remains very basic to justify a publication in a journal as Sustainability. 

The idea is interesting, but the authors must perform robust research to add value to their paper. 

Parts of the response letter, I am referring: 

Comment 1 

The theme of the paper is interesting;   however, my main concern is regarding the contributions of the findings to the literature.   The analysis provided is too simple to justify the publication.   Maybe authors could perform a survey or a Delphi study with experts in the area to develop a route to be followed.   Or compare the situation of different countries. 

Answers to comment 1: 

Thank you very much for referees’ reports.   We seriously thought about the reviewer's opinion and answered the question.   Because of the epidemic, many cities have adopted strict prevention and control policies, including the cities in the study area.   Under such conditions, it is very difficult for us to carry out the survey, which is a pity in the research.   Regarding to ‘compare the situation of different countries’, due to the differences in statistical caliber, it is difficult to make comparisons between countries.    The comments of reviewers are very valuable and enlightening to me.   Based on the comment, we have made the following adjustments to the Limitations: 

Comment 2 

Why the analysis is up to 2019 and not up to 2021 or 2022?   Authors must update the data or justify the reason for using this period in the text.   Indeed, the authors even mention the covid-19 pandemic that became a worldwide problem in 2020. 

Answers to comment 2: 

Thank you very much for referees’ reports.   We seriously thought about the reviewer's opinion and answered the question.   The impact of urban human settlements on residents' health has a lagging effect, which is mainly manifested in the form that after the change of urban human settlements, it will not have an immediate effect on residents' health, and its impact will not disappear in a short period of time.   In the next few years, it will also have a significant effect on residents' health.   the covid-19 pandemic that became a worldwide problem in 2020, therefore, it is necessary to evaluate the health of the human settlements before the epidemic.

Author Response

Dear editors and Reviewers:

Thank you very much for your review and comments on the manuscript. We are honored to see that you think this study has very important practical significance. All the questions you mentioned are important for us to improve the quality of the paper, and all your questions have been seriously considered and answered in detail. We hope this satisfies readers' needs for these content.

Here, we attached the revised manuscript in the editable words for your approval. A document answering every question from the referees was also summarized and enclosed. If you have any questions, please contact us without hesitation.

Reviewer #2:

Comment 1

The theme of the paper is interesting; however, my main concern is regarding the contributions of the findings to the literature. The analysis provided is too simple to justify the publication. Maybe authors could perform a survey or a Delphi study with experts in the area to develop a route to be followed. Or compare the situation of different countries.

Answers to comment 1:

Thank you very much for referees’ reports. We seriously thought about the reviewer's opinion and answered the question. We carefully read the relevant literature and learned the practices of different countries. Their experiences can give us useful inspiration. we have made the following adjustments to the Recommendations.

Comment 2

Why the analysis is up to 2019 and not up to 2021 or 2022? Authors must update the data or justify the reason for using this period in the text. Indeed, the authors even mention the covid-19 pandemic that became a worldwide problem in 2020.

Answers to comment 2:

Thank you very much for referees’ reports. We seriously thought about the reviewer's opinion and updated the data to 2020.

Due to too many modifications, please see the attachment.
